# AN EMPIRICAL STUDY ON EVALUATION METRICS OF GENERATIVE ADVERSARIAL NETWORKS

## ABSTRACT

Despite the widespread interest in generative adversarial networks (GANs), few works have studied the metrics that quantitatively evaluate GANs' performance. In this paper, we revisit several representative sample-based evaluation metrics for GANs, and address the important problem of *how to evaluate the evaluation metrics*. We start with a few necessary conditions for metrics to produce meaningful scores, such as distinguishing real from generated samples, identifying mode dropping and mode collapsing, and detecting overfitting. Then with a series of carefully designed experiments, we are able to comprehensively investigate existing sample-based metrics and identify their strengths and limitations in practical settings. Based on these results, we observe that kernel Maximum Mean Discrepancy (MMD) and the 1-Nearest-Neighbour (1-NN) two-sample test seem to satisfy most of the desirable properties, *provided that the distances between samples are computed in a suitable feature space*. Our experiments also unveil interesting properties about the behavior of several popular GAN models, such as whether they are memorizing training samples, and how far these state-of-the-art GANs are from perfect.

## 1 INTRODUCTION

Generative adversarial networks (GANs) (Goodfellow et al., 2014) have been studied extensively in recent years. Besides producing surprisingly plausible images of faces (Radford et al., 2015; Larsen et al., 2015) and bedrooms (Radford et al., 2015; Arjovsky et al., 2017; Gulrajani et al., 2017), they have also been innovatively applied in, for example, semi-supervised learning (Odena, 2016; Makhzani et al., 2015), image-to-image translation (Isola et al., 2016; Zhu et al., 2017), and simulated image refinement (Shrivastava et al., 2016). However, despite the availability of a plethora of GAN models (Arjovsky et al., 2017; Qi, 2017; Radford et al., 2015; Zhao et al., 2016), their evaluation is still predominantly qualitative, very often resorting to manual inspection of the visual fidelity of generated images. Such evaluation is time-consuming, subjective and possibly misleading. Given the inherent limitations of *qualitative* evaluations, proper *quantitative* metrics are crucial for the development of GANs to avoid the human factors and guide the design of better models.

Possibly the most popular metric is the Inception Score (Salimans et al., 2016), which measures the quality and diversity of the generated images using an external model, the Google Inception network (Szegedy et al., 2014), trained on the large scale ImageNet dataset (Deng et al., 2009). Some other metrics are less widely used but still very valuable. Wu et al. (2016) proposed a sampling method to estimate the log-likelihood of generative models, by assuming a Gaussian observation model with a fixed variance. Bounliphone et al. (2015) propose to use maximum mean discrepancies (MMDs) for model selection in generative models. Lopez-Paz & Oquab (2016) apply the classifier two-sample test, a well-studied tool in statistics, to assess the difference between the generated and target distribution.

Although these evaluation metrics are shown to be effective on various tasks, it is unclear in which scenarios their scores are meaningful, and in which other scenarios, prone to misinterpretations. Given that evaluating GANs is already challenging, it can only be more difficult to evaluate the evaluation metrics themselves. Most existing works attempt to justify their proposed metrics by showing a strong correlation with human evaluation (Salimans et al., 2016; Lopez-Paz & Oquab, 2016). However, human evaluation tends to be biased towards the visual quality of generated samples and neglect the overall distributional characteristics, which are arguably just as important for unsupervised learning.

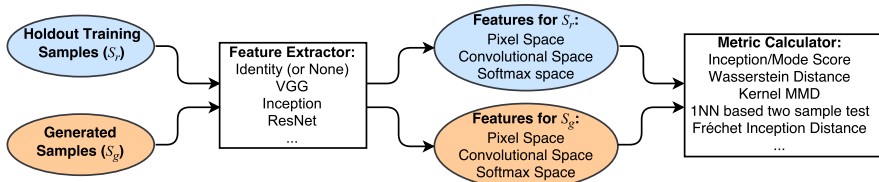

Figure 1: A schematic layout of the typical approach for sample based GAN evaluation methods.

In this paper we comprehensively examine the existing literature on sample-based quantitative evaluation of GANs. We address the challenge of evaluating the metrics themselves by carefully designing a series of experiments, through which we hope to answer the following important questions: 1.) What are reasonable characterizations of the behavior of existing sample-based metrics for GANs? 2.) What are the strengths and limitations of these metrics? 3.) Which metrics are preferred accordingly? 4.) How are the metrics helpful in understanding and improving GANs?

Ultimately, we hope that this paper will establish good principles on choosing, applying, interpreting and designing evaluation metrics for GANs in practical settings. We will also release the source code for all experiments and metrics examined, providing the community with off-the-shelf tools to debug and improve their GAN algorithms.

## 2 BACKGROUND

We briefly review the original GAN framework proposed by Goodfellow et al. (2014). Description of the GAN variants used in our experiments is deferred to the Appendix A.

### 2.1 GENERATIVE ADVERSARIAL NETWORKS

Let $\mathcal{X} = \mathbb{R}^{d \times d}$ be the space of natural images. Given i.i.d. samples $S_r = \{\mathbf{x}_1^r, \ldots, \mathbf{x}_n^r\}$ drawn from a real distribution $\mathbb{P}_r$ over $\mathcal{X}$, we would like to learn a parameterized distribution $\mathbb{P}_g$ that approximates the distribution $\mathbb{P}_r$.

The setup of generative adversarial networks is as follows. We define two networks, the discriminator $D : \mathcal{X} \to [0, 1]$ and the generator $G : \mathcal{Z} \to \mathcal{X}$, where $\mathcal{Z}$ is some latent space. Given a distribution $\mathbb{P}_z$ over $\mathcal{Z}$ (usually an isotropic Gaussian), the distribution $\mathbb{P}_g$ is defined as $G(\mathbb{P}_z)$. Optimization is performed with respect to a joint loss for $D$ and $G$

$$\min_G \max_D L(D, G) = \mathbb{E}_{\mathbf{x} \sim \mathbb{P}_r} \log \left[ D(\mathbf{x}) \right] + \mathbb{E}_{\mathbf{z} \sim \mathbb{P}_z} \left[ \log(1 - D(G(\mathbf{z}))) \right]. \tag{1}$$

Intuitively, the discriminator $D$ outputs a probability for every $\mathbf{x} \in \mathcal{X}$ that corresponds to its likelihood of being drawn from $\mathbb{P}_r$, and the loss function encourages the generator $G$ to produce samples that maximize this probability. Practically, the loss is approximated with finite samples from $\mathbb{P}_r$ and $\mathbb{P}_g$, and optimized with alternating steps for $D$ and $G$ using gradient descent.

To evaluate the generator, we would like to design a metric $\rho$ that measures the "dissimilarity" between $\mathbb{P}_g$ to $\mathbb{P}_r$.[1] In theory, with both distributions known, common choices of $\rho$ include the Kullback-Leibler divergence (KLD), Jensen-Shannon divergence (JSD) and total variation. However, in practical scenarios, $\mathbb{P}_r$ is unknown and only the finite samples in $S_r$ are observed. Furthermore, it is almost always intractable to compute the exact density of $\mathbb{P}_g$, but much easier to sample $S_g = \{\mathbf{x}_1^g, \ldots, \mathbf{x}_m^g\} \sim \mathbb{P}_g^m$ (especially so for GANs). Given these limitations, we focus on *empirical* measures $\hat{\rho} : \mathcal{X}^n \times \mathcal{X}^m \to \mathbb{R}$ of "dissimilarity" between samples from two distributions.

### 2.2 SAMPLE BASED METRICS

We mainly focus on sample based evaluation metrics that follow a common setup illustrated in Figure 1. The *metric calculator* is the key element, for which we briefly introduce five representative methods: Inception Score (Salimans et al., 2016), Mode Score (Che et al., 2016) , Kernel MMD (Gretton et al., 2007), Wasserstein distance, Fréchet Inception Distance (FID) (Heusel et al., 2017), and 1-nearest neighbor (1-NN)-based two sample test (Lopez-Paz & Oquab, 2016). All of them are model agnostic and require only finite samples from the generator.

---

[1]Note that $\rho$ does not need satisfy symmetry or triangle inequality, so it is not, mathematically speaking, a distance metric between $\mathbb{P}_g$ and $\mathbb{P}_r$. We still call it a metric throughout this paper for simplicity.

**The Inception Score** is arguably the most widely adopted metric in the literature. It uses a image classification model $\mathcal{M}$, the Google Inception network (Szegedy et al., 2016), pre-trained on the ImageNet (Deng et al., 2009) dataset, to compute

$$\text{IS}(\mathbb{P}_g) = e^{\mathbb{E}_{\mathbf{x} \sim \mathbb{P}_g}[KL(p_{\mathcal{M}}(y|\mathbf{x})||p_{\mathcal{M}}(y))]}, \tag{2}$$

where $p_{\mathcal{M}}(y|\mathbf{x})$ denotes the label distribution of $\mathbf{x}$ as predicted by $\mathcal{M}$, and $p_{\mathcal{M}}(y) = \int_{\mathbf{x}} p_{\mathcal{M}}(y|\mathbf{x}) \, d\mathbb{P}_g$, i.e. the marginal of $p_{\mathcal{M}}(y|\mathbf{x})$ over the probability measure $\mathbb{P}_g$. The expectation and the integral in $p_{\mathcal{M}}(y|\mathbf{x})$ can be approximated with i.i.d. samples from $\mathbb{P}_g$. A higher IS has $p_{\mathcal{M}}(y|\mathbf{x})$ close to a point mass, which happens when the Inception network is very confident that the image belongs to a particular ImageNet category, and has $p_{\mathcal{M}}(y)$ close to uniform, i.e. all categories are equally represented. This suggests that the generative model has both high quality and diversity. Salimans et al. (2016) show that the Inception Score has a reasonable correlation with human judgment of image quality. We would like to highlight two specific properties: 1) the distributions on both sides of the KL are dependent on $\mathcal{M}$, and 2) the distribution of the real data $\mathbb{P}_r$, or even samples thereof, are not used anywhere.

**The Mode Score** is an improved version of the Inception Score. Formally, it is given by

$$\text{MS}(\mathbb{P}_g) = e^{\mathbb{E}_{\mathbf{x} \sim \mathbb{P}_g}[KL(p_{\mathcal{M}}(y|\mathbf{x})||p_{\mathcal{M}}(y))] - KL(p_{\mathcal{M}}(y)||p_{\mathcal{M}}(y^*))}, \tag{3}$$

where $p_{\mathcal{M}}(y^*) = \int_{\mathbf{x}} p_{\mathcal{M}}(y|\mathbf{x}) \, d\mathbb{P}_r$ is the marginal label distribution for the samples from the real data distribution. Unlike the Inception Score, it is able to measure the dissimilarity between the real distribution $\mathbb{P}_r$ and generated distribution $\mathbb{P}_g$ through the term $KL(p_{\mathcal{M}}(y)||p_{\mathcal{M}}(y^*))$.

**The Kernel MMD** (Maximum Mean Discrepancy), defined as

$$\text{MMD}(\mathbb{P}_r, \mathbb{P}_g) = \left( \mathbb{E}_{\substack{\mathbf{x}_r, \mathbf{x}_r' \sim \mathbb{P}_r, \\ \mathbf{x}_g, \mathbf{x}_g' \sim \mathbb{P}_g}} \left[ k(\mathbf{x}_r, \mathbf{x}_r') - 2k(\mathbf{x}_r, \mathbf{x}_g) + k(\mathbf{x}_g, \mathbf{x}_g') \right] \right)^{\frac{1}{2}}, \tag{4}$$

measures the dissimilarity between $\mathbb{P}_r$ and $\mathbb{P}_g$ for some fixed kernel function $k$. Given two sets of samples from $\mathbb{P}_r$ and $\mathbb{P}_g$, the empirical MMD between the two distributions can be computed with finite sample approximation of the expectation. A lower MMD means that $\mathbb{P}_g$ is closer to $\mathbb{P}_r$. The Parzen window estimate (Gretton et al., 2007) can be viewed as a specialization of Kernel MMD.

**The Wasserstein distance** between $\mathbb{P}_r$ and $\mathbb{P}_g$ is defined as

$$\text{WD}(\mathbb{P}_r, \mathbb{P}_g) = \inf_{\gamma \in \Gamma(\mathbb{P}_r, \mathbb{P}_g)} \mathbb{E}_{(\mathbf{x}^r, \mathbf{x}^g) \sim \gamma} \left[ d(\mathbf{x}^r, \mathbf{x}^g) \right], \tag{5}$$

where $\Gamma(\mathbb{P}_r, \mathbb{P}_g)$ denotes the set of all joint distributions (i.e. probabilistic couplings) whose marginals are respectively $\mathbb{P}_r$ and $\mathbb{P}_g$, and $d(\mathbf{x}^r, \mathbf{x}^g)$ denotes the base distance between the two samples. For discrete distributions with densities $p_r$ and $p_g$, the Wasserstein distance is often referred to as the Earth Mover's Distance (EMD), and corresponds to the solution to the optimal transport problem

$$\text{WD}(p_r, p_g) = \min_{w \in \mathbb{R}^{n \times m}} \sum_{i=1}^{n} \sum_{j=1}^{m} w_{ij} d(\mathbf{x}_i^r, \mathbf{x}_j^g) \quad \text{s.t.} \quad \sum_{j=1}^{m} w_{i,j} = p_r(\mathbf{x}_i^r) \; \forall i, \sum_{i=1}^{n} w_{i,j} = p_g(\mathbf{x}_j^g) \; \forall j. \tag{6}$$

This is the finite sample approximation of $\text{WD}(\mathbb{P}_r, \mathbb{P}_g)$ used in practice. Similar to MMD, the Wasserstein distance is lower when two distributions are more similar.

**The Fréchet Inception Distance (FID)** was recently introduced by Heusel et al. (2017) to evaluate GANs. Formally, it is given by

$$\text{FID}(\mathbb{P}_r, \mathbb{P}_g) = \|\mu_r - \mu_g\| + \text{Tr}(\mathbf{C}_r + \mathbf{C}_g - 2(\mathbf{C}_r \mathbf{C}_g)^{1/2}), \tag{7}$$

where $\mu_r$ ($\mu_g$) and $\mathbf{C}_r$ ($\mathbf{C}_g$) are the mean and covariance of the real (generated) distribution, respectively. Note that under the Gaussian assumption on both $\mathbb{P}_r$ and $\mathbb{P}_g$, the Fréchet distance is equivalent to the Wasserstein-2 distance.

**The 1-Nearest Neighbor classifier** is used in two-sample tests to assess whether two distributions are identical. Given two sets of samples $S_r \sim \mathbb{P}_r^n$ and $S_g \sim \mathbb{P}_g^m$, with $|S_r| = |S_g|$, one can compute the leave-one-out (LOO) accuracy of a 1-NN classifier trained on $S_r$ and $S_g$ with positive labels for $S_r$ and negative labels for $S_g$. Different from the most common use of accuracy, here the 1-NN

classifier should yield a $\sim 50\%$ LOO accuracy when $|S_r| = |S_g|$ is large. This is achieved when the two distributions match. The LOO accuracy can be lower than $50\%$, which happens when the GAN overfits $\mathbb{P}_g$ to $S_r$. In the (hypothetical) extreme case, if the GAN were to memorize every sample in $S_r$ and re-generate it exactly, i.e. $S_g = S_r$, the accuracy would be $0\%$, as every sample from $S_r$ would have it nearest neighbour from $S_g$ with zero distance. The 1-NN classifier belongs to the two-sample test family, for which any binary classifier can be adopted in principle. We will only consider the 1-NN classifier because it requires no special training and little hyperparameter tuning.

Lopez-Paz & Oquab (2016) considered the 1-NN accuracy primarily as a statistic for two-sample testing. In fact, it is more informative to analyze it for the two classes separately. For example, a typical outcome of GANs is that for both real *and* generated images, the majority of their nearest neighbors are generated images due to mode collapse. In this case, the LOO 1-NN accuracy of the real images would be relatively low (desired): the mode(s) of the real distribution are usually well captured by the generative model, so a majority of real samples from $S_r$ are surrounded by generated samples from $S_g$, leading to low LOO accuracy; whereas the LOO accuracy of the generated images is high (not desired): generative samples tend to collapse to a few mode centers, thus they are surrounded by samples from the same class, leading to high LOO accuracy. For the rest of the paper, we distinguish these two cases as 1-NN accuracy (real) and 1-NN accuracy (fake).

### 2.3 OTHER METRICS

All of the metrics above are, what we refer to as "*model agnostic*": they use the generator as a black box to sample the generated images $S_g$. Model agnostic metrics should not require a density estimation from the model. We choose to only experiment with model agnostic metrics, which allow us to support as many generative models as possible for evaluation without modification to their structure. We will briefly mention some other evaluation metrics not included in our experiments.

Kernel density estimation (KDE, or Parzen window estimation) is a well-studied method for estimating the density function of a distribution from samples. For a probability kernel $K$ (most often an isotropic Gaussian) and i.i.d samples $\mathbf{x}_1, \ldots, \mathbf{x}_n$, we can define the density function at $\mathbf{x}$ as $p(\mathbf{x}) \approx \frac{1}{z} \sum_{i=1}^n K(\mathbf{x} - \mathbf{x}_i)$, where $z$ is a normalizing constant. This allows the use of classical metrics such as KLD and JSD. However, despite the widespread adoption of this technique to various applications, its suitability to estimating the density of $\mathbb{P}_r$ or $\mathbb{P}_g$ for GANs has been questioned by Theis et al. (2015) since the probability kernel depends on the Euclidean distance between images.

More recently, Wu et al. (2016) applied annealed importance sampling (AIS) to estimate the marginal distribution $p(\mathbf{x})$ of a generative model. This method is most natural for models that define a conditional distribution $p(\mathbf{x}|\mathbf{z})$ where $\mathbf{z}$ is the latent code, which is not satisfied by most GAN models. Nevertheless, AIS has been applied to GAN evaluation by assuming a Gaussian observation model. We exclude this method from our experiments as it needs the access to the generative model to compute the likelihood, instead of only depending on a finite sample set $S_g$.

## 3 EXPERIMENTS WITH GAN EVALUATION METRICS

### 3.1 FEATURE SPACE

All the metrics introduced in the previous section, except for the Inception Score and Mode Score, access the samples $\mathbf{x}$ only through pair-wise distances. The Kernel MMD requires a fixed kernel function $k$, typically set to an isotopic Gaussian; the Wasserstein distance and 1-NN accuracy use the underlying distance metric $d$ directly; all of these methods are highly sensitive to the choice that distance.

It is well-established that pixel representations of images do not induce meaningful Euclidean distances (Forsyth & Ponce, 2011). Small translations, rotations, or changes in illumination can increase distances dramatically with little effect on the image content. To quantity the similarity between distributions of images, it is therefore desirable to use distances invariant to such transformations. The choice of distance function can be re-interpreted as a choice of representation, by defining the distance in a more general form as $d(\mathbf{x}, \mathbf{x}') = \|\phi(\mathbf{x}) - \phi(\mathbf{x}')\|_2$, where $\phi(\cdot)$ is some general mapping of the input into a semantically meaningful feature space. For Kernel MMD, this corresponds to computing the usual inner product in the feature space $\phi(\mathcal{X})$.

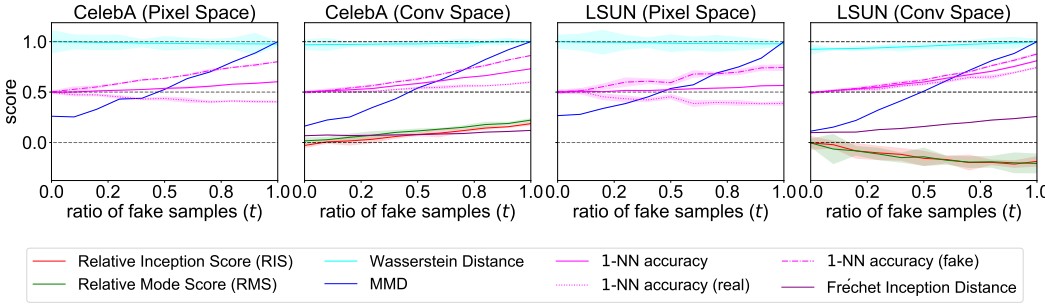

Figure 2: Distinguishing a set of real images (in the training set) from a mixed set of real images and GAN generated images. For the metric to be discriminative, its score should increase as the fraction of generated samples in the mix increases. RIS and RMS fail as they decrease with the fraction of generated samples in $S_g$ on LSUN. Wasserstein and 1-NN accuracy (real) fail in pixel space as they do not increase.

Inspired by recent works from Upchurch et al. (2017); Larsen et al. (2015) which show that convolutional neural networks may linearize the image manifold, we propose to operate in the feature space of an external model pre-trained on the ImageNet dataset. For efficiency, we use a 34-layer ResNet[2] as the feature extractor. Our experiments show that other models such as VGG or Inception give very similar results.

To illustrate our point, we show failure examples of the pixel space distance for evaluating GANs in this section, and highlight that using a proper feature space is key to obtaining meaningful results when applying the distance-based metrics. The usage of a well-suited feature space enables us to draw more optimistic conclusions on GAN evaluation metrics than in Theis et al. (2015).

## 3.2 SETUP

For the rest of this major section, we introduce what in our opinion are necessary conditions for good metrics for GANs. After the introduction of each condition, we use it as a criterion to judge the effectiveness of the metrics presented in Section 2, through carefully designed empirical experiments.

The experiments are performed on two standard benchmark datasets for generative models, CelebA[3] and LSUN bedrooms[4]. To remove the degree of freedom induced by feature representation, we use the Inception Score (IS) and Mode Score (MS) computed from the softmax probabilities of the same ResNet-34 model as the other metrics, instead of the Inception model. We also compute the Inception Score over the real training data $S_r$ as an upper bound, which we denote as $IS_0$. Moreover, to be consistent with other metrics where lower values correspond to better models, we report the *relative inverse* Inception Score $RIS = (1 - IS/IS_0)$ in tables and plots, after computing IS the Inception Score. We similarly report the relative inverse Mode Score (RMS). Although RIS and RMS operate in the softmax space, we always compare them together with other metrics in the convolutional space for simplicity. For all the plots in this paper, shaded areas denote the standard deviations, computed by running the same experiment 5 times with different random seeds.

## 3.3 DISCRIMINABILITY

**Mixing of generated images.** Arguably, the most important property of a metric $\hat{\rho}$ for measuring GANs is the ability to distinguish generated images from real images. To test this property, we sample a set $S_r$ consisting of $n$ ($n = 2000$ if not otherwise specified) real images uniformly from the training set, and a set $S_g(t)$ of the same size $n$ consisting of a *mix* of real samples and generated images from a DCGAN (Radford et al., 2015) trained on the same training set, where $t$ denotes the ratio of generated images.

The computed values of various metrics between $S_r$ and $S_g(t)$ for $t \in [0, 1]$ are shown in Figure 2. Since $\mathbb{P}_r$ should serve as a lower bound for any metric, we expect that any reasonable $\hat{\rho}$ should increase as the ratio of generated images increases. This is indeed satisfied for all the metrics *except*:

---

[2]We use the official pre-trained ResNet-34 model from PyTorch `http://tinyurl.com/pytorch`.

[3]CelebA is a large-scale high resolution face dataset with more than 200,000 centered celebrity images.

[4]LSUN consists of around one million images for each of 10 scene classes. Following standard practice, we only take the *bedroom* scene.

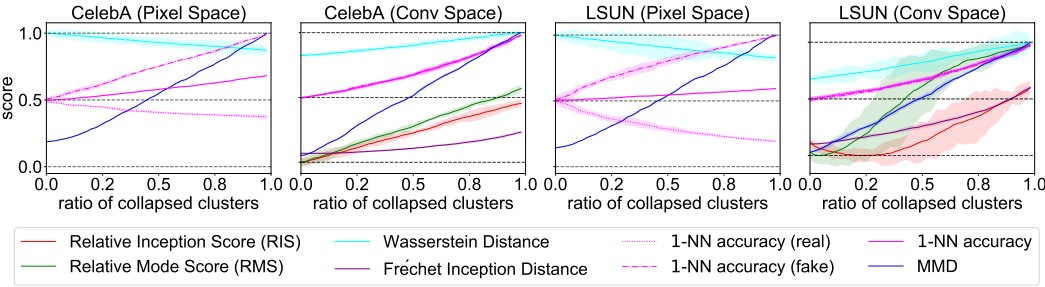

Figure 3: Experiment on simulated mode collapsing. A metric score should increase to reflect the mismatch between true distribution and generated distribution as more modes are collapsed towards their cluster center. All metrics respond correctly in convolutional space. In pixel space, both Wasserstein distance and 1-NN accuracy (real) fail as they decrease in response to more collapsed clusters.

1) RIS and RMS (red and green curves) on LSUN, which decrease as more fake samples are in the mix; 2) 1-NN accuracy of real samples (dotted magenta curve) computed in *pixel space*, which also appears to be a decreasing function; and 3) Wasserstein Distance (cyan curve), which almost remains unchanged when $t$ varies.

The reason that RIS and RMS do not work well here is likely because they are not suitable for images beyond the ImageNet categories. Although other metrics operate in the convolutional feature space also depend on a network pretrained on ImageNet, the convolutional features are much more general than the specific softmax representation. The failure of Wasserstein Distance is possibly due to an insufficient number of samples, which we will discuss in more detail when we analyze the sample efficiency of various metrics in a latter subsection. The last paragraph of Section 2.2 explains why the 1-NN accuracy for real samples (dotted magenta curve) is always lower than that for generated samples (dashed magenta curve). In the pixel space, more than half of the samples from $S_r$ have the nearest neighbor from $S_g(t)$, indicating that the DCGAN is able to represent the modes in the pixel space quite well.

We also conducted the same experiment using 1) random noise images and 2) images from an entirely different distribution (e.g. CIFAR-10), instead of DCGAN generated images to construct $S_g(t)$. We call these injected samples as *out-of-domain violations* since they are not in $\mathcal{X}$, the domain of the real images. These settings yield similar results as in Figure 2, thus we omit their plots.

**Mode collapsing and mode dropping.** In realistic settings, $\mathbb{P}_r$ is usually very diverse since natural images are inherently multimodal. Many have conjectured that $\mathbb{P}_g$ differs from $\mathbb{P}_r$ by *reducing diversity*, possibly due to the lack of model capacity or inadequate optimization (Arora et al., 2017). This is often manifested itself for generative models in a mix of two ways: *mode dropping*, where some hard-to-represent modes of $\mathbb{P}_r$ are simply "ignored" by $\mathbb{P}_g$; and *mode collapsing*, where several modes of $\mathbb{P}_r$ are "averaged" by $\mathbb{P}_g$ into a single mode, possibly located at a midpoint. An ideal metric should be sensitive to these two phenomena.

To test for *mode collapsing*, we first randomly sample both $S_r$ and $S'_r$ as two disjoint sets of 2000 *real* images. Next, we find 50 clusters in the whole training set with $k$-means and progressively replace each cluster by its respective cluster center to *simulate* mode collapse. Figure 3 shows computed values of $\hat{\rho}(S_r, S'_r)$ as the number of replaced (collapsed) clusters, denoted as $C$ increases. Ideally, we expect the scores increase as $C$ grows. We first observe that all the metrics are able to respond correctly when distances are computed in the *convolutional feature space*. However, the *Wasserstein* metric (cyan curve) breaks down in pixel space, as it considers a collapsed sample set (with $C > 0$) being closer to the real sample set than another set of real images (with $C = 0$). Moreover, although the overall 1-NN accuracy (solid magenta curve) follows the desired trend, the real and fake parts follow opposite trends: 1-NN *real* accuracy (dotted magenta curve) decreases while 1-NN *fake* accuracy (dashed magenta curve) increases. Again, this is inline with our explanation given in the last paragraph of Section 2.2.

To test for *mode dropping*, we take $S_r$ as above and construct $S'_r$ by randomly removing clusters. To keep the size of $S'_r$ constant, we replace images from the removed cluster with images randomly selected from the remaining clusters. Figure 4 shows how different metrics react to the number of removed clusters, also denoted as $C$. All scores effectively discriminate against mode dropping

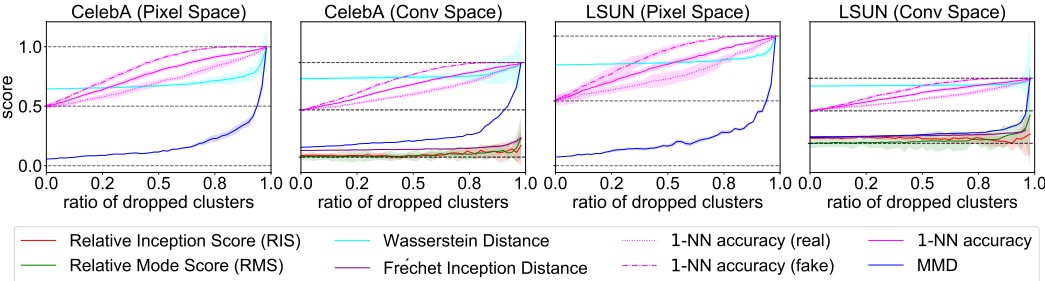

Figure 4: Experiment on simulated mode dropping. A metric score should increase to reflect the mismatch between true distribution and generated distribution as more modes are dropped. All metrics except RIS and RMS respond correctly, as they only increase slightly in value even when almost all modes are dropped.

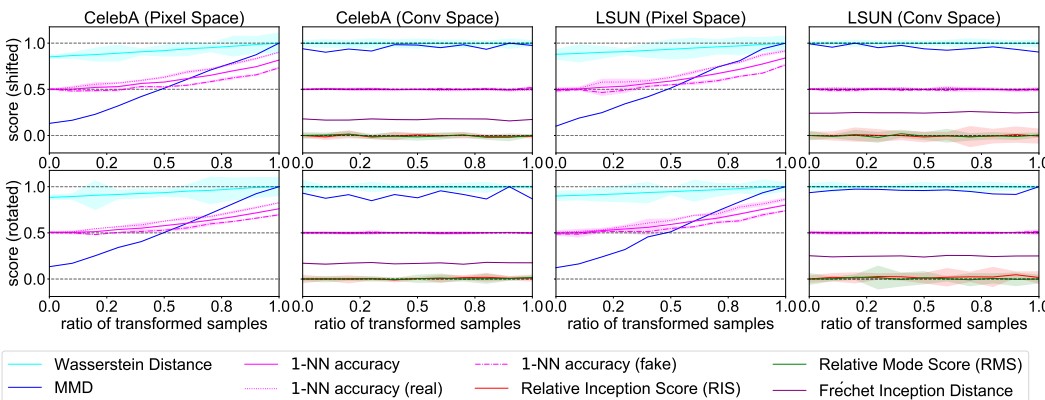

Figure 5: Experiment on robustness of each metric to small transformations (rotations and translations). All metrics should remain constant across all mixes of real and transformed real samples, since the transformations do not alter semantics of the image. All metrics respond correctly in convolutional space, but behave incorrectly in pixel space. This experiment illustrates the unsuitability of distances in pixel space.

*except the RIS and RMS* - they remain almost indifferent when some modes are dropped. Again, this is perhaps caused by the fact that the Inception/Mode Score were originally designed for datasets with classes overlapping with the ImageNet dataset, and they do not generalize well to other datasets.

### 3.4 ROBUSTNESS TO TRANSFORMATIONS

GANs are widely used for image datasets, which have the property that certain transformations to the input do not change its semantic meaning. Thus an ideal evaluation metric should be invariant to such transformations to some extent. For example, a generator trained on CelebA should not be penalized by a metric if its generated faces are shifted by a few pixels or rotated by a small angle.

Figure 5 shows how the various metrics react to such small transformation to the images. In this experiment, $S_r$ and $S_r'$ are two disjoint sets of 2000 *real* images sampled from the training data. However, a proportion of images from $S_r'$ are randomly shifted (up to 4 pixels) or rotated (up to 15 degrees). We can observe from the results that metrics operating in the convolutional space (or softmax space for RIS and RMS) are robust to these transformations, as all the curves are approximated horizontal as the ratio of transformed samples increases. This is not that surprising as

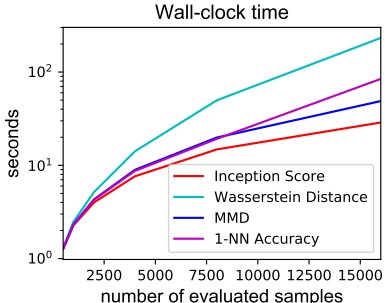

Figure 7: Measurement of wall-clock time for computing various metrics as a function of the number of samples. All metrics are practical to compute for a sample of size 2000, but Wasserstein distance does not scale to large sample sizes.

convolutional networks are well know for being invariant to certain transformations (Mallat, 2016). In comparison, in the pixel space all the metrics consider the shifted/rotated images as drawn from a different distribution, highlighting the importance of computing distances in a proper feature space.

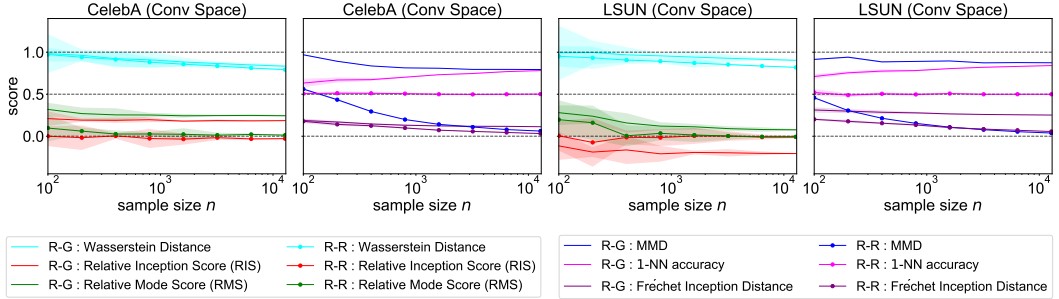

Figure 6: The score of various metrics as a function of the number of samples. An ideal metric should result in a large gap between the real-real (R-R; $\hat{\rho}(S_r, S'_r)$) and real-fake (R-G; $\hat{\rho}(S_r, S_g)$) curves in order to distinguish between real and fake distributions using as few samples as possible. Compared with Wasserstein distance, MMD and 1-NN accuracy require much fewer samples to discriminate real and generated images, while RIS totally fails on LSUN as it scores generated images even better (lower) than real images.

### 3.5 EFFICIENCY

A practical GAN evaluation metric should be able to compute "accurate" scores from a *reasonable number* of samples and within an affordable computation cost, such that it can be computed, for example, after each training epoch to monitor the training process.

**Sample efficiency.** Here we measure the sample efficiency of various metrics by investigating how many samples are needed for each of them in order to discriminate a set of generated samples $S_g$ (from DCGAN) from a set of real samples $S'_r$. To do this, we introduce a reference set $S_r$, which is also uniformly sampled from the real training data, but is disjoint with $S'_r$. All three sample sets have the same size, i.e., $|S_r| = |S'_r| = |S_g| = n$. We expect that an ideal metric $\rho$ should correctly score $\hat{\rho}(S_r, S'_r)$ lower than $\hat{\rho}(S_r, S_g)$ with a relatively small $n$. In other words, the number of samples $n$ needed for the metric to distinguish $S'_r$ and $S_g$ can be viewed as its sample complexity.

In Figure 6 we show the individual scores as a function of $n$. We can observe that MMD, FID and 1-NN accuracy computed in convolution feature space are able to distinguish the two set of images $S_g$ (solid blue and magenta curves) and $S'_r$ (dotted solid blue and magenta curves) with relatively few samples. The Wasserstein distance (cyan curves) is not discriminative with samples size less than 1000, while the RIS even considers the generated samples to be more "real" than the real samples on the LSUN dataset (the red curves in the third panel). The dotted lines in Figure 6 also quantify how fast the scores converge to their expectations as we increase the sample size. Note that MMD for $\hat{\rho}(S_r, S'_r)$ converges very quickly to zero and gives discriminative scores with few samples, making it a practical metric for comparing GAN models.

**Computational efficiency.** Fast computation of the empirical metric is of practical concern as it helps researchers monitor the training process and diagnose problems early on, or perform early stopping. In Figure 7 we investigate the computational efficiency of the above metrics by showing the wall-clock time (in log scale) to compute them as a function of the number of samples. For a typical number of 2000 samples, it only takes about 8 seconds to compute each of these metrics on an NVIDIA TitanX GPU. In fact, the majority of time is spent on extracting features from the ResNet model. Only the Wasserstein distance becomes prohibitively slow for large sample sizes.

### 3.6 DETECTING OVERFITTING

Overfitting is an artifact of training with finite samples. If a GAN successfully memorizes the training images, i.e., $\mathbb{P}_g$ is a uniform distribution over the training sample set $S_r^{tr}$, then the generated samples $S_g$ becomes a uniformly drawn set of $n$ samples from $S_r^{tr}$, and any reasonable $\hat{\rho}$ should be close to 0. The Wasserstein distance, MMD and 1-NN two sample test are able to detect overfitting in the following sense: if we *hold out* a validation set $S_r^{val}$, then $\hat{\rho}(S_g, S_r^{val})$ should be significantly higher than $\hat{\rho}(S_g, S_r^{tr})$ when $\mathbb{P}_g$ memorizes a part of $S_r^{tr}$. The difference between them can informally be viewed as a form of "generalization gap".

We simulate the overfitting process by defining $S'_r$ as a mix of samples from the training set $S_r^{tr}$ and a second holdout set, disjoint from both $S_r^{tr}$ and $S_r^{val}$. Figure 8 shows the gap $\hat{\rho}(S_g, S_r^{val}) - \hat{\rho}(S_g, S_r^{tr})$ of the various metrics as a function of the overlapping ratio between $S'_r$ and $S_r^{tr}$. The left most point

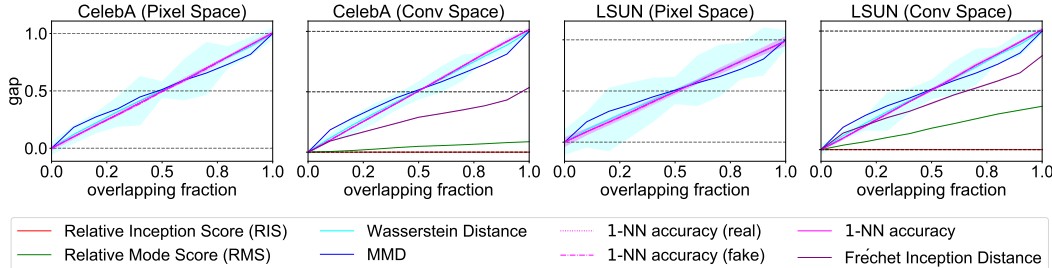

Figure 8: Experiment on detecting overfitting of generated samples. As more generated samples overlap with real samples from the training set, the gap between validation and training score should increase to signal overfitting. All metrics behave correctly except for RIS and RMS, as these two metrics do not increase when the fraction of overlapping samples increases.

of each curve can be viewed as the score $\hat{\rho}(S'_r, S_r^{val})$ computed on a validation set since the overlap ratio is 0. For better visualization, we normalize the Wasserstein distance and MMD by dividing their corresponding score when $S'_r$ and $S_r$ have no overlap. As shown in Figure 8, all the metrics except RIS and RMS reflect that the "generalization gap" increases as $S'_r$ overfits more to $S_r$. The failure of RIS is not surprising: it totally ignores the real data distribution as we discussed in Section 2.2. While the reason that RMS also fails to detect overfitting may again be its lack of generalization to datasets with classes not contained in the ImageNet dataset. In addition, RMS operates in the softmax space, the features in which might be too specific compared to the features in the convolutional space.

## 4    DISCUSSIONS AND CONCLUSION

Based on the above analysis, we can summarize the advantages and inherent limitations of the five evaluation metrics, and conditions under which they produce meaningful results. With some of the metrics, we are able to study the problem of overfitting (see Appendix C), perform model selection on GAN models and compare GAN models without resorting to human evaluation based on cherry-picked samples (see Appendix D).

**The Inception Score** does show a reasonable correlation with the quality and diversity of generated images, which explains the wide usage in practice. However, it is ill-posed mostly because it only evaluates $\mathbb{P}_g$ as an image generation model rather than its similarity to $\mathbb{P}_r$. Blunt violations like mixing in natural images from an entirely different distribution completely deceives the Inception Score. As a result, it may encourage the models to simply learn sharp and diversified images (or even some adversarial noise), instead of $\mathbb{P}_r$. This also applies to the **Mode Score**. Moreover, the Inception Score is unable to detect overfitting since it cannot make use of a holdout validation set.

**Kernel MMD** works surprising well when it operates in the feature space of a pre-trained ResNet. It is always able to identify generative/noise images from real images, and both its sample complexity and computational complexity are low. Given these advantages, even though MMD is biased, we recommend its use in practice.

**Wasserstein distance** works well when the base distance is computed in a suitable feature space. However, it has a high sample complexity, a fact that has been independently observed by (Arora et al., 2017). Another key weakness is that computing the exact Wasserstein distance has a time complexity of $O(n^3)$, which is prohibitively expensive as sample size increases. Compared to other methods, Wasserstein distance is less appealing as a *practical evaluation metric*.

**Fréchet Inception Distance** performs well in terms of discriminability, robustness and efficiency. It appears to be a good evaluation metric for GANs, even it only takes into consideration the first two order moments of the distributions.

**1-NN classifier** seems to be an ideal metric for evaluating GANs. Not only does it enjoy all the advantages of the other metrics, it also outputs a score in the interval $[0, 1]$, similar to the accuracy/error in classification problems. When the generative distribution perfectly match the true distribution, perfect score (i.e., $50\%$ accuracy) is attainable with a reasonable (e.g., 1000) sample size. From Figure 2, we find that typical GAN models tend to achieve lower LOO accuracy for *real* samples (*1-NN accuracy (real)*), while higher LOO accuracy for *generated* samples (*1-NN accuracy (fake)*). This suggests that GANs are able to capture *modes* from the training distribution, such that the

majority of training samples distributed around the mode centers have their nearest neighbor from the generated images, yet most of the generated images are still surrounded by generated images as they are collapsed. The observation indicates that the *mode collapse* problem is prevalent for typical GAN models. We also note that this problem, however, cannot be effectively detected by human evaluation or the widely used Inception Score.

Overall, our empirical study suggests that the choice of feature space in which to compute various metrics is crucial. In the convolutional space of a ResNet pretrained on ImageNet, both MMD and 1-NN accuracy appear to be good metrics in terms of discriminability, robustness and efficiency. Wasserstein distance has very poor sample efficiency, while Inception Score and Mode Score appear to be unsuitable for datasets that are very different from ImageNet. We will release our source code for all these metrics, providing researchers with an off-the-shelf tool to compare and improve GAN algorithms.

Based on the two most prominent metrics, MMD and 1-NN accuracy, we study the overfitting problem of DCGAN and WGAN (in Appendix C). Despite the widespread belief that GANs are overfitting to the training data, we find that this does not occur unless there are very few training samples. This raises an interesting question regarding the generalization of GANs in comparison to the supervised setting. We hope that future work can contribute to explaining this phenomenon.

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

## A  GAN VARIANTS USED IN OUR EXPERIMENTS

Many GAN variants have been proposed recently. In this paper we consider several of them, which we briefly review in this section.

**DCGAN** (Radford et al., 2015). The generator of a DCGAN takes a lower dimensional input from a uniform noise distribution, then projects and reshapes it to a small convolutional representation with many feature maps. After applying a series of four fractionally-strided convolutions, the generator converts this representation into a $64 \times 64$ pixel image. DCGAN is optimized by minimizing the Jensen-Shannon divergence between the real and generated images.

**WGAN** (Arjovsky et al., 2017). A critic network that outputs unconstrained real values is used in place of the discriminator. When the critic is Lipschitz, this network approximates the Wasserstein distance between $S_r$ and $S_g$. A Lipschitz condition is enforced by clipping the critic networks' parameters to stay within a predefined bounding box.

**WGAN with gradient penalty** (Gulrajani et al., 2017) improves upon WGAN by enforcing the Lipschitz condition with a gradient penalty term. This method significantly improves the convergence speed and the quality of the images generated by a WGAN.

**LSGAN** (Mao et al., 2016). Least Squares GAN adopts the least squares loss function instead of the commonly used sigmoid cross entropy loss for the discriminator, essentially minimizing the Pearson $\chi^2$ divergence between the real distribution $\mathbb{P}_r$ and generative distribution $\mathbb{P}_g$.

## B  THE CHOICE OF FEATURE SPACE

The choice of features space is crucial for all these metrics. Here we consider several alternatives to the convolutional features from the 34-layer ResNet trained on ImageNet. In particular, we compute various metrics using the features extracted by (1) the VGG and Inception networks; (2) a 34-layer ResNet with random weights; (3) a ResNet classifier trained on the same dataset as the GAN models. We use the features extracted from these models to test all metrics in the discriminative experiments we performed in Section 3. All experimental settings are identical except for the third experiments, which is performed on the CIFAR-10 dataset (Krizhevsky & Hinton, 2009) instead as we need class labels to train the classifier. Note that we consider setting (3) only for analytical purposes. It is not a practical choice as GANs are mainly designed for unsupervised learning and we should not assume the existence of ground truth labels.

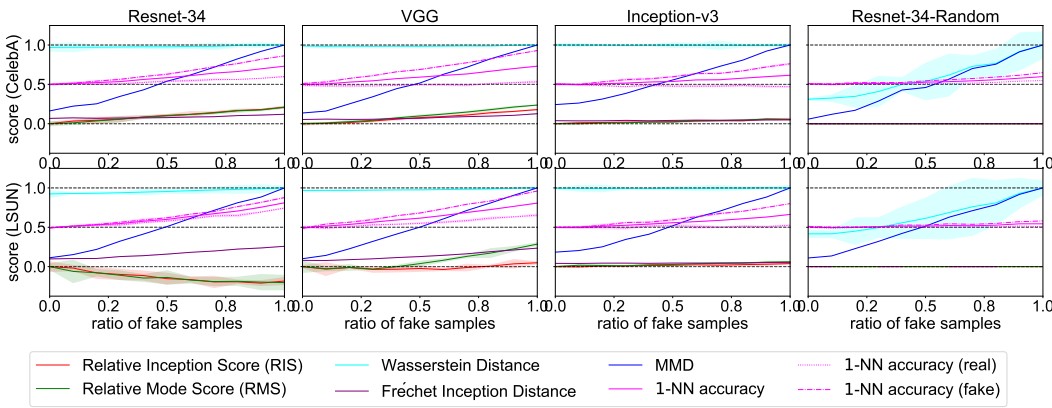

Figure 9: Comparison of all metrics in different feature spaces. When using different trained networks, the trends of all metrics are very similar. Most metrics work well even in a random network, but Wasserstein distance has very high variance and the magnitude of increase for 1-NN accuracy is small.

The results are shown in Figure 9 and Figure 10, from which several observations can be made: (1) switching from ResNet-34 to VGG or Inception has little effect to the metric scores; (2) the features from a random network still works for MMD, while it makes the Wasserstein distance unstable and 1-NN accuracy less discriminative. Not surprisingly, the Inception Score and Mode Score becomes meaningless if we use the softmax values from the random network; (3) features extracted from the classifier trained on the same dataset as the GAN model also offers high discriminability for these

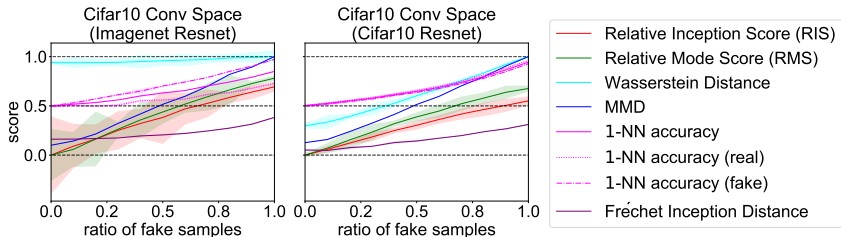

Figure 10: Using features extracted from a ResNet trained on CIFAR-10 (right plot) to evaluate a GAN model trained on the same dataset. Compared to using an extractor trained on ImageNet, the metrics appear to have lower variance. However, this may due to the feature dimensionality being smaller for CIFAR-10.

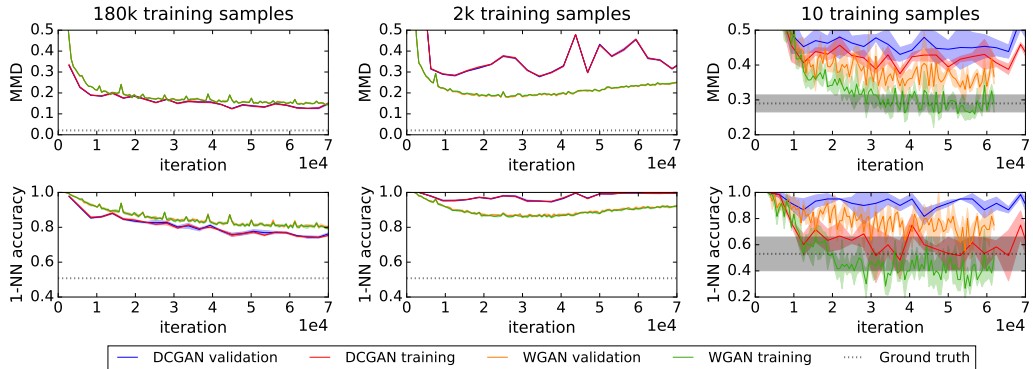

Figure 11: Training curves of DCGAN and WGAN on a large (*left* two panels), small (*middle* two panels) and tiny (*right* two panels) subsets of CelebA. Note that for the first four plots, blue (yellow) curves almost overlap with the red (green) curves, indicating no overfitting detected by the two metrics. Overfitting only observed on the tiny training set, with MMD score and 1-NN accuracy significantly worse (higher) on the validation set.

metrics, especially for the Wasserstein distance. However, this may be simply due to the fact that the feature dimensionality of the ResNet trained on CIFAR-10 is much smaller than that of the ResNet-34 trained on ImageNet (64 v.s. 512).

## C ARE GANS OVERFITTING TO THE TRAINING DATA?

We trained two representative GAN models, DCGAN (Radford et al., 2015) and WGAN (Arjovsky et al., 2017) on the CelebA dataset. Out of the ∼200,000 images in total, we holdout 20,000 images for validation, and the rest for training. As the training set is sufficiently large, which makes overfitting unlikely to occur, we also create a small training set and a tiny training set respectively with only 2000 and 10 images sampled from the full training set.

The training setting for DCGAN and WGAN strictly follow their original implementation, except that we change the default number of training iterations such that both models are sufficiently updated. For each metric, we compute their score on 2000 real samples and 2000 generated samples, where the real samples are drawn from either the training set or the validation set, giving rise to training and validation scores. The results are shown in Figure 11, from which we can make several observations:

- The training and validation scores almost overlap with each other with 2000 or 180k training samples, showing that both DCGAN and WGAN do not overfit to the training data under of these metrics. Even when using only 2000 training samples, there is still no significant difference between the training score and validation score. This shows that the training process of GANs behaves quite differently from those of supervised deep learning models, where a model can easily achieve 0 training error while behaving like random guess on the validation set (Zhang et al., 2016). [5]

---

[5]We observed that even memorizing 50 images is difficult for GAN models.

Table 1: Comparison of several GAN models on the LSUN dataset

|  |  | Real | DCGAN | WGAN | WGAN-GP | LSGAN |
|---|---|---|---|---|---|---|
| Conv Space | MMD | 0.019 | 0.205 | 0.270 | **0.194** | 0.232 |
|  | 1-NN Accuracy | 0.499 | 0.825 | 0.920 | **0.812** | 0.871 |
|  | 1-NN Accuracy (real) | 0.495 | **0.759** | 0.880 | 0.765 | 0.804 |
|  | 1-NN Accuracy (fake) | 0.503 | 0.892 | 0.961 | **0.860** | 0.938 |

- DCGAN outperforms WGAN on the full training set under both metrics, and converges faster. However, WGAN is much more stable on the small training set, and converges to better positions.

# D   COMPARISON OF POPULAR GAN MODELS BASED ON QUANTITATIVE EVALUATION METRICS

Based on our analysis, we chose MMD and 1-NN accuracy in the feature space of a 34-layer ResNet trained on ImageNet to compare several state-of-the-art GAN models. All scores are computed using 2000 samples from the holdout set and 2000 generated samples. The GAN models evaluated include DCGAN (Radford et al., 2015), WGAN (Arjovsky et al., 2017), WGAN with gradient penalty (WGAN-GP ) (Gulrajani et al., 2017), and LSGAN (Mao et al., 2016) , all trained on the CelebA dataset. The results are reported in Table 1, from which we highlight three observations:

- WGAN-GP performs the best under most of the metrics.

- DCGAN achieves 0.759 overall 1-NN accuracy on real samples, slightly better than 0.765 achieved by WGAN-GP; while the 1-NN accuracy on generated (fake) samples achieved by DCGAN is higher than that by WGAN-GP (0.892 *v.s.* 0.860). This seems to suggest that DCGAN is better at capturing modes in the training data distribution, while its generated samples are more collapsed compared to WGAN-GP. Such subtle difference is unlikely to be discovered by the Inception Score or human evaluation.

- The 1-NN accuracy for all evaluated GAN models are higher than $0.8$ , far above the ground truth of $0.5$. The MMD score of the four GAN models are also much larger than that of ground truth (0.019). This indicates that even state-of-the-art GAN models are far from learning the true distribution.

