# OpenReview forum: "An empirical study on evaluation metrics of generative adversarial networks"
_ICLR.cc/2018/Conference — Reject_

### Official Review · AnonReviewer3 · 2017-11-23
**A good survey of GAN evaluation metrics with exhaustive experimental evaluations.**

**Rating:** 8
**Confidence:** 3

**Review:**

In the paper, the authors discuss several GAN evaluation metrics.
Specifically, the authors pointed out some desirable properties that GANS evaluation metrics should satisfy.
For those properties raised, the authors experimentally evaluated whether existing metrics satisfy those properties or not.
Section 4 summarizes the results, which concluded that the Kernel MMD and 1-NN classifier in the feature space are so far recommended metrics to be used.

I think this paper tackles an interesting and important problem, what metrics are preferred for evaluating GANs.
In particular, the authors showed that Inception Score, which is one of the most popular metric, is actually not preferred for several reasons.
The result, comparing data distributions and the distribution of the generator would be the preferred choice (that can be attained by Kernel MMD and 1-NN classifier), seems to be reasonable.
This would not be a surprising result as the ultimate goal of GAN is mimicking the data distribution.
However, the result is supported by exhaustive experiments making the result highly convincing.

Overall, I think this paper is worthy for acceptance as several GAN methods are proposed and good evaluation metrics are needed for further improvements of the research field.

---

> ### Author Response · Authors · 2017-12-23
> **thanks**
>
> Thanks a lot for the nice summary of the paper, and the positive comments!

---

### Official Review · AnonReviewer1 · 2017-11-28
**Evaluation of GAN evaluation metrics**

**Rating:** 7
**Confidence:** 4

**Review:**

Thanks for an interesting paper.

The paper evaluates popular GAN evaluation metrics to better understand their properties. The "novelty" of this paper is a bit hard to assess. However, I found their empirical evaluation and experimental observations to be very interesting. If the authors release their code as promised, the off-the-shelf tool would be a very valuable contribution to the GAN community.

In addition to existing metrics, it would be useful to add Frechet Inception Distance (FID) and Multi-scale structural similarity (MS-SSIM).

Have you considered approximations to Wasserstein distance? E.g. Danihelka et al proposed using an independent Wasserstein critic to evaluate GANs:
Comparison of Maximum Likelihood and GAN-based training of Real NVPs
https://arxiv.org/pdf/1705.05263.pdf

How sensitive are the results to hyperparameters? It would be interesting to see some sensitivity analysis as well as understand the correlations between different metrics for different hyperparameters (cf. Appendix G in https://arxiv.org/pdf/1706.04987.pdf)

Do you think it would be useful to compare other generative models (e.g. VAEs) using these evaluation metrics? Some of the metrics don't capture perceptual similarity, but I'm curious to hear what you think.

---

> ### Author Response · Authors · 2017-12-23
> **Response**
>
> Thanks for your positive comments! We will definitely release our code as promised.
>
>
> #FID and MS-SSM#
> Thanks for pointing us to these interesting works. We have updated our paper and included FID. MS-SSIM does not fit as well into the evaluation framework we are considering as it is an approach that can be combined with the metrics we studied, instead of being a new metric to be compared with.
>
>
> #Approximations to Wasserstein distance#
> Thanks for the suggestion. We have indeed considered it, but have found that even the exact Wasserstein distance is not an appealing metric. This makes us believe that it is probably not worth to consider an approximated version. In addition, the approximated Wasserstein distance involves extra hyperparameters, such as network configurations and optimization settings, which further complicates the matter.
>
>
> #Sensitivity to hyperparameters#
> A good evaluation metric should be robust in terms of hyperparameter setting, or ideally, have no hyperparameters. This is indeed satisfied by all the metrics we investigated in this paper, except the MMD. MMD has a single hyperparameter, the Gaussian kernel width, which we empirically set to the averaged pairwise distance of the training data. It appears that MMD is quite insensitive to this hyperparameter.
>
> #Evaluate other generative models and perceptual similarity#
> In principle it should be possible to use these metrics to evaluate other generative models,  but we have not really investigated that further. Our metrics should work if the goal of other generative models is also approximating the data distribution.

---

### Official Review · AnonReviewer2 · 2017-12-01
**An empirical comparison of 4 metrics for GANs**

**Rating:** 5
**Confidence:** 5

**Review:**

The paper describes an empirical evaluation of some of the most common metrics to evaluate GANs (inception score, mode score, kernel MMD, Wasserstein distance and LOO accuracy).

The paper is well written, clear, organized and easy to follow.

Given that the underlying application is image generation, the authors move from a pixel representation of images to using the feature representation given by a pre-trained ResNet, which is key in their results and further comparisons. They analyzed discriminability, mode collapsing and dropping, robustness to transformations, efficiency and overfitting.

Although this work and its results are very useful for practitioners, it lacks in two aspects. First, it only considers a single task for which GANs are very popular. Second, it could benefit from a deeper (maybe theoretical analysis) of some of the questions. Some of the conclusions could be further clarified with additional experiments (e.g., Sec 3.6 ‘while the reason that RMS also fails to detect overfitting may again be its lack of generalization to datasets with classes not contained in the ImageNet dataset’).

---

> ### Author Response · Authors · 2017-12-23
> **Response**
>
> Thank you for your comments!
>
>
> #only considers a single task#
> We focus on the image generation task because, as you pointed out, GANs are most popular in this context. The systematic approach proposed to investigate GAN evaluation metrics is however quite general. For example, we introduce how to evaluate a metric by checking the discriminability between generated images and real images, the sample efficiency, the sensitivity to mode dropping and mode collapsing, etc.
>
> # it could benefit from deeper (maybe theoretical) analysis of some of the questions#
> We totally agree that a deep understanding of our obtained results is beneficial, and we did provide possible explanations whenever possible. However, our focus in this paper is primarily to identify the properties of an ideal GAN evaluation metric, and how to explicitly and empirically investigate the strengths and limitations of a given metric.
>
> We are diving into more depth for some of the issues that surfaced in this process, for example why a specific metric works well or fails in terms of discriminability. Hopefully, these could inspires future work on further analyzing existing metrics or designing better ones.
>
> Previously, evaluation metrics for GANs were proposed and applied - often without detailed investigation. We claim that evaluation metrics themselves need to be carefully evaluated first.

---

### Official Review · AnonReviewer4 · 2017-12-06
**An empirical comparison of metrics for GAN**

**Rating:** 5
**Confidence:** 3

**Review:**

This paper introduces a comparison between several approaches for evaluating GANs. The authors consider the setting of a pre-trained image models as generic representations of generated and real images to be compared. They compare the evaluation methods based on five criteria termed disciminability, mode collapsing and mode dropping, sample efficiency,computation efficiency, and robustness to transformation. This paper has some interesting insights and a few ideas of how to validate an evaluation method. The topic is an important one and a very difficult one. However, the work has some problems in rigor and justification and the conclusions are overstated in my view.

Pros
-Several interesting ideas for evaluating evaluation metrics are proposed
-The authors tackle a very challenging subject

Cons
-It is not clear why GANs are the only generative model considered
-Unprecedented visual quality as compared to other generative models has brought the GAN to prominence and yet this is not really a big factor in this paper.
-The evaluations rely on using a pre-trained imagenet model as a representation. The authors point out that different architectures yield similar results for their analysis, however it is not clear how the biases of the learned representations affect the results. The use of learned representations needs more rigorous justification
-The evaluation for discriminative metric, increased score when mix of real and unreal increases, is interesting but it is not convincing as the sole evaluation for “discriminativeness” and seems like something that can be gamed.
- The authors implicitly contradict the argument of Theis et al against monolithic evaluation metrics for generative models, but this is not strongly supported.

Several references I suggest:
https://arxiv.org/abs/1706.08500 (FID score)
https://arxiv.org/abs/1511.04581 (MMD as evaluation)

---

> ### Author Response · Authors · 2017-12-23
> **response**
>
> Thanks for your comments.
>
> #Why only GANs#
> We only consider GANs because adding another generative model type would necessarily add another dimension to the comparison and complicate it further. GANs are very popular and widely used, so we hope there is a sufficient amount of interest even if we restrict ourselves to this domain. It should be interesting to extend our research to other generative models in the future.
>
>
> #Visual quality#
> Our paper focuses on evaluating the metrics for GAN, instead of evaluating the generated images of GANs. Indeed, visual quality of GAN generated images are unprecedented, but the ultimate goal of GAN is to learn the hidden generative distribution. From this perspective, GANs can still be improved in various aspects. For example, mode collapse and mode drop cannot be easily discovered by visual inspection only, but can be evaluated using our proposed methods.
>
>
> #Using pretrained model as feature extractor#
> The pretrained models on ImageNet are very general and robust. They have been widely used for transfer learning (e.g., object detection, semantic segmentation), image style transfer, image super-resolution, etc. The widely used Inception Score also relies on the Inception model pretrained on ImageNet, and it works quite well in practice as an GAN evaluation metric.  In addition, our experiments (Figure 9 in appendix) show that all the observations we have on ResNet also hold on VGG and Inception.
>
>
> #Sole evaluation discriminability#
> It is important to emphasize that discriminating real and unreal images is not the sole evaluation for “discriminativeness” in our paper. We also considered the prevalent mode dropping and mode collapsing problems. Moreover, even for unreal images, we experimented with three different settings: 1) images generated by a GAN; 2) random noise and 3) images from an entirely different distribution.
>
> Discriminativeness is not a sufficient condition for a good metric, but seems to be a necessary condition. If a metric does not even pass the discriminativeness test, or other tests in our paper, it might not be a good metric.
>
> #Contradict the argument of Theis et al#
> We would like to argue that our observations are inline with what have been observed by Theis et al. Specifically, if we directly compute the distances in the pixel space as in Theis et al, most the metrics would fail in terms of discriminability, which is also been observed in their paper. We are able to draw more optimistic observations because of the introduction of a proper feature space.
>
>
> #References#
> Thanks for point us to these papers. We have cited both papers, and updated our results with the FID score included.

---

### Public Comment · (anonymous) · 2017-11-14
**Fréchet Inception Distance (FID) for evaluating GANs**

[1] proposed the Fréchet Inception Distance (FID) for evaluating GANs which is not mentioned here. To gain a broader insight into evaluation metrics of GANs the authors should also discuss this quality measure.

[1] https://arxiv.org/abs/1706.08500

---

> ### Public Comment · (anonymous) · 2017-11-22
> **Shameless**
>
> This "review" appears to be self-promotion.

---

> > ### Public Comment · (anonymous) · 2017-11-24
> > **IMHO**
> >
> > This review, in particular it's a public comment, makes a valid point. A paper titled "An empirical study on evaluation metrics of generative adversarial networks" should consider all evaluation metrics out there or at least give reasons as to why some were not considered.

---

> > > ### Public Comment · (anonymous) · 2017-11-26
> > > **Mentioning all of missing measures**
> > >
> > > If this were true then why not mention "all" of the potential metrics that are missing from the paper and instead only cite one paper that is missing.
> > >
> > > Other works not covered or mentioned as a potential metric:
> > >
> > > Revisiting Classifier Two-Sample Tests
> > > https://arxiv.org/abs/1610.06545
> > >
> > > GENERATIVE ADVERARIAL METRIC
> > > https://openreview.net/pdf?id=wVqzLo88YsG0qV7mtLq7
> > >
> > > Mode Regularized Generative Adversarial Networks
> > > https://arxiv.org/abs/1612.02136
> > > (They introduce the mode score as a modification to the inception score)
> > >
> > > Again the original "review" only mentioned one particular paper for perhaps obvious reasons.

---

> ### Author Response · Authors · 2017-11-26
> **Preliminary Results and Thoughts**
>
> Thanks for pointing this paper to us. The FID is essentially a distance metric for probability distributions, thus it fits into the evaluation framework we investigated. We will include it in our updated version.
>
> Following [1], we make the Gaussian distribution assumption on the real/generated data, and test the discriminability of FID. Our preliminary results show that it behaves similarly to the Wasserstein distance under this test. For other properties, we speculate that FID will have a better time/sample complexity than the Wasserstein distance due to the simplified Gaussian distribution assumption. But it might be less sensitive to mode collapse.
>
> Our paper aims to provide a framework to investigate different properties of GAN evaluation metrics. Thus researchers can use it to analyze any sample-based evaluation metric that fits into the framework. As there exist many metrics for probability distributions, we can only focus on several most typical ones (especially those already been used by the GAN community) in our paper.

---

> ### Author Response · Authors · 2018-01-22
> **Update with FID results**
>
> We have updated our paper with FID results included. The FID score does appear to be an appealing metric according to our criterions.

---

### Public Comment · (anonymous) · 2017-11-24
**Questions**

1) This may not be a very relevant question. But from your current knowledge of evaluation metrics of GANs, what kind of evaluation metrics are relevant for using for other GAN tasks such as unpaired translation (e.g. CycleGAN)?

2) Can you say anything with evaluation metrics for text generation?

3) I'm also curious as to why FID was omitted, and I'd like to know which one would be better, FID or 1-NN, and in addition the sample efficiency of FID in particular.

---

> ### Author Response · Authors · 2017-11-26
> **Reply**
>
> 1) This is an interesting question. For CycleGAN, the mode dropping/collapsing problem might be less several due to the reconstruction loss. Therefore, we may prefer those metrics, e.g., MMD, that have better discriminability (between generated distribution and target distribution).
>
> 2) The critical part might be how to define the distance between two sentences/documents. Once we have a well-defined distance, all the metrics investigated in this paper can be applied. The word mover’s distance (WMD) [1] appears to be an ideal candidate.
> [1] Kusner et al, From Word Embeddings To Document Distances, ICML, 2015
>
> 3) We plan to include FID in the updated version. Please refer to our reply to the other comments for some preliminary results.

---

### Public Comment · (anonymous) · 2018-01-26
**Question: extension to Conditional GANs**

Can we apply these metrics to evaluate conditional versions of GANs or it needs some specific adaptations?
What are the most relevant metrics for evaluating conditional GANs?

---

### Decision · Program_Chairs · 2018-01-29
**ICLR 2018 Conference Acceptance Decision**

**Decision:**

Reject

**Comment:**

The problem addressed here is an important one: What is a good evaluation metric for generative models?  A good selection of popular metrics are analyzed for their appropriateness for model selection of GANs.  Two popular approaches are recommended: the kernel Maximum Mean Discrepancy (MMD) and the 1-Nearest-Neighbour (1-NN) two-sample test.  This seems reasonable, but the present work was not recommended for acceptance by 2 reviewers who raised valid concerns.

From a readability perspective, it would be nice to simply list the answer to question (1) directly in the introduction.  One must read more than a few pages to get to the answer of why the metrics that are advocated were picked.  It need not read like a mystery.

R4: "The evaluations rely on using a pre-trained imagenet model as a representation. The authors point out that different architectures yield similar results for their analysis, however it is not clear how the biases of the learned representations affect the results. The use of learned representations needs more rigorous justification"

R2: "First, it only considers a single task for which GANs are very popular. Second, it could benefit from a deeper (maybe theoretical analysis) of some of the questions." - the first point of which is also related to a concern of R4.

Given the overall high selectivity of ICLR, the present submission falls short.